# Systemic Lipid Metabolism Dysregulation as a Possible Driving Force of Fracture Non-Unions?

**DOI:** 10.3390/bioengineering11111135

**Published:** 2024-11-11

**Authors:** Lovorka Grgurević, Ruđer Novak, Lucija Jambrošić, Marko Močibob, Morana Jaganjac, Mirna Halasz, Grgur Salai, Stela Hrkač, Milan Milošević, Tomislav Vlahović, Jeronim Romić, Dražen Matičić, Dinko Vidović

**Affiliations:** 1Center for Translational and Clinical Research, Department of Proteomics, School of Medicine, University of Zagreb, 10000 Zagreb, Croatia; rudjer.novak@mef.hr (R.N.);; 2Department of Anatomy, “Drago Perovic”, School of Medicine, University of Zagreb, 10000 Zagreb, Croatia; 3Biomedical Research Center Salata, School of Medicine, University of Zagreb, 10000 Zagreb, Croatia; 4Department of Chemistry, Faculty of Science, University of Zagreb, 10000 Zagreb, Croatia; 5Laboratory for Oxidative Stress, Division of Molecular Medicine, Ruder Boskovic Institute, 10000 Zagreb, Croatia; 6Department of Pulmonology, University Hospital Dubrava, 10000 Zagreb, Croatia; 7Department of Clinical Immunology, Allergology and Rheumatology, University Hospital Dubrava, 10000 Zagreb, Croatia; 8Department for Environmental and Occupational Health, Andrija Stampar School of Public Health, School of Medicine, University of Zagreb, 10000 Zagreb, Croatia; 9Clinic of Traumatology, University Hospital Center “Sestre Milosrdnice”, 10000 Zagreb, Croatia; 10Clinics for Surgery, Orthopedics and Ophthalmology, Faculty of Veterinary Medicine, University of Zagreb, 10000 Zagreb, Croatia; 11School of Dental Medicine, University of Zagreb, 10000 Zagreb, Croatia; 12Department of Surgery, School of Medicine, University of Zagreb, 10000 Zagreb, Croatia

**Keywords:** bone healing, bone fracture, non-union, proteomics, metabolomics

## Abstract

Introduction: Non-unions are fractures that do not heal properly, resulting in a false joint formation at the fracture site. This condition leads to major health issues and imposes a burden on national healthcare systems. The etiology of non-unions is still not fully understood; therefore, we aimed to identify potential systemic factors that may contribute to their formation. Materials and methods: We conducted a cross-sectional concomitant proteomic and metabolomic pilot study of blood plasma in patients with non-unions (N = 11) and compared them with patients with bone fracture in the normal active healing phase (N = 12). Results: We found five significantly upregulated proteins in the non-union group: immunoglobulin heavy variable 3–74, immunoglobulin lambda variable 2–18, low-density lipoprotein receptor-related protein 4, zinc-alpha-2-glycoprotein, and serum amyloid A-1 protein; and we found one downregulated protein: cystatin-C. The metabolomic study found differences in alanine, aspartate and glutamate metabolism pathways between two groups. Conclusions: The combined results of proteomic and metabolomic analyses suggest that the dysregulation of lipid metabolism may contribute to non-union formation.

## 1. Introduction

Bone fractures pose a major public health challenge because they lead to work incapacity, disability, and increased healthcare costs, particularly in patients with osteoporosis. Primary bone healing leads to intramembranous ossification as mesenchymal cells differentiate to osteoblasts and start to deposit new bone tissue without the formation of bone callus or inflammation. The secondary mechanism is the most common type of bone fracture healing, and includes endochondral ossification, where cartilage is deposed by osteoblasts as an intermediate before ossification [1]. As blood accumulates at the fracture site, the healing process starts with hematoma formation, which serves as an initial scaffold for bone repair. The successive inflammatory processes trigger the proliferation of chondrocytes. Blood coagulum is then replaced with soft callus, made from fibrous tissue and cartilage, followed by hard callus formation. The final phase of the healing process is bone remodeling [2]. It depends on the interaction between osteoblast and osteoclast precursor cells [3]. Osteoblasts are bone-forming cells that differentiate from stem cells in the bone marrow and periosteum, in a process orchestrated by bone morphogenic proteins (BMPs), which represent the most researched group of proteins in the bone healing process [4]. BMPs are part of the transforming growth factor β family that play an important role in embryogenesis and early development, and in the adult organism, they support fracture healing and vasculature remodeling [5,6]. RhBMP2 and BMP7 have both shown the strongest initiation of cartilage and bone development and have thus been approved for clinical use. However, the US Food and Drug Administration (FDA) discontinued BMP7 due to safety and efficacy issues and more potent alternatives such as BMP6, which was found to be more potent in the promotion of osteoblast differentiation [7,8]. RhBMP2, on the other hand, is currently being studied as a graft material in fusion procedures aiming to correct spinal deformities [9].

Most fractures heal without complications, but in 5–10% of cases, the healing process is prolonged or abnormal, leading to a condition known as non-union fracture or pseudoarthrosis [10]. The definition of non-union (pseudoarthrosis) is under debate: the Food and Drug Administration (FDA) suggests that 9 months of failed bone union after fracture, or a lack of healing progress for 3 months is sufficient for the diagnosis, while some clinicians further prolong this to 6 months. Finally, some propose radiological imaging as the only source of definitive diagnosis, where the obliteration of the medullary canal of long bones, termination of physiological reaction to fracture, or interposition of cartilaginous or fibrous tissue is found. The symptoms of non-unions include decreased mobility and chronic pain and are a problem in orthopedics and traumatology that requires significant efforts to alleviate pain and restore function in patients [11,12]. The etiology of non-unions is not completely explained. Non-union commonly develops in unstable fractures which are biologically active, oligotrophic or hypertrophic. Inactive non-unions result from a compromised blood supply due to blood supply damage from periosteal stripping, or, it can occur if there is bone necrosis or defect. In some cases, non-unions occur due to both, an instability and poor blood supply. Although efforts are being made to identify potential non-union biomarkers and disease modulators, currently, there are no universally accepted guidelines for recommended procedures or treatment options. This gap highlights the need for standardization and consensus on best practices for diagnosis and treatment [13]. It was shown that an increased vascularity of humeral non-unions indicates better patient outcomes, sparking the “diamond concept” theory [11,14] regarding the benefits of osteoconductive scaffolds, bone growth factors, and osteogenic cell in long bone non-union therapy. This theory emphasizes the importance of these elements and their interaction as a necessary step in adequate fracture healing. Some have proposed that the use of platelet-rich plasma (PRP), mesenchymal stem cells (MSCs) or bone morphogenetic proteins (BMPs) can help reduce healing time and increase union rates [14,15,16]. Prolonged use of proton pump inhibitors was correlated with greater risk of tibial and femoral shaft fracture non-unions. As these medications are one of the most frequently used drugs worldwide, their potential influence on non-unions might be significant. This phenomenon can be explained by the reduced solubility of calcium in the higher pH levels of the stomach, leading to lower calcium absorption and increased PTH activity, which in turn stimulates higher osteoclastic activity, particularly in older populations [17]. Additionally, it was shown on a rabbit femoral fracture model that the prolonged use of COX-2 inhibitors (ibuprofen and rofecoxib) reduced healing speed and increased non-union rates [18]. Despite these efforts, the mechanisms underlying the occurrence of non-unions are still unknown, and specific studies need to be conducted for their better understanding.

In this study, we aimed to examine the systemic differences in molecular profiles between non-unions and fractures in the early stages of healing. We compared blood samples in two groups of patients with similar characteristics but different outcomes in bone healing using plasma proteomics and plasma metabolomics in order to detect crucial proteins, metabolites as well as underlying biological processes that might be responsible for impairments in bone healing.

## 2. Materials and Methods

### 2.1. Study Participants and Study Outline

This cross-sectional observational study was approved by the Institutional Ethics Committee of the Sestre Milosrdnice University Hospital Center Zagreb (EP-003-06/20–03/023). Subjects were enrolled between April 2022 and September 2023 at the Sestre Milosrdnice University Hospital Centre Zagreb, where they also provided a signed informed consent. All fractures were radiologically verified and confirmed by three experienced orthopedic surgeons.

### 2.2. Plasma Proteomics

Patient blood samples were drawn into 3.8% sodium citrate tubes (anticoagulant/blood ratio of 1/9), centrifuged at 15 min at 4 °C and 3000× *g* to obtain plasma, and stored at −80 °C until analysis. Before use, individual samples were thawed and centrifuged at 16,000× *g* for 10 min, using the clarified supernatant in subsequent experiments. Total protein concentration was determined in each sample using the RC DC Lowry protein assay (BioRad, Hercules, CA, USA) according to the manufacturer’s instructions. Samples containing 50 μg protein per patient were transferred to 10 kDa centrifugal filter units for further processing. Briefly, proteins were denatured in 8 M urea, alkylated in 55 mM iodoacetamide (in 8 M urea), and finally digested overnight in 25 mM ammonium bicarbonate with 1 μg of TPCK-treated trypsin (Worthington Industries, Columbus, OH, USA). The obtained tryptic peptides were desalted and concentrated using in-house made Stage Tips mini-columns [19]. Tryptic peptides were then separated by HPLC on an Easy-nLC 1200 System (Thermo Fisher Scientific, Waltham, MA, USA) in a gradient of acetonitrile in 0.1% formic acid on a PepMap C18 25-centimeter-long nano-column. Peptides were sequenced on a Q Exactive Plus (Thermo Fisher Scientific, Waltham, MA, USA) mass spectrometer, set up to measure automated cycles consisting of full MS scans and MS/MS scans of the ten most intense ions. Full MS scans were taken using the Orbitrap analyzer at an *m*/*z* range of 300 to 1800, and a resolution of 70,000, with an internal calibration of the mass spectrometer using the lock mass setting.

### 2.3. Plasma Metabolomics

All the chemicals used for metabolomic analyses were purchased from Sigma-Aldrich Chemical Co. (St. Louis, MO, USA), except for N,O-bis(trimethylsilyl) trifluoroacetamide (BSTFA) with 1% trimethylchlorosilane (TMCS) (Pierce Chemical Co., Rockford, IL, USA), FAME mix (Supelco, Bellefonte, PA, USA), and API-TOF reference mass solution kit (Agilent Technologies, Waldbronn, Germany). All solvents were of LC–MS grade. Sample preparation and metabolite extraction were carried out as described in [20]. Quality controls were prepared by pooling and mixing equal volumes of each sample (10 μL). They were processed along with the rest of the samples. The liquid chromatography–electrospray ionization–quadrupole time-of-flight mass spectrometry (LC-ESI-QTOF-MS) and gas chromatography–electron ionization–quadrupole mass spectrometry (GC-EI-Q-MS) analytical platforms were used as previously described [20]. Briefly, LC-MS analysis was performed with an Agilent Technologies Series 1200 binary solvent delivery system (Agilent Technologies, Waldbronn, Germany), coupled to an Agilent 6520 Accurate-Mass Q-TOF detector. The Agilent 7890A gas chromatograph coupled to an inert MSD with Quadrupole (Agilent Technologies 5975, Agilent Technologies, Waldbronn, Germany) was applied for GC-MS analysis. The results obtained from both platforms were combined to analyze the metabolic profile of the plasma samples.

### 2.4. Data Analysis–Participants Characteristics

Participants’ characteristics were presented in the form of a table. Categorical variables are presented as a number (N) and, where applicable, as a percentage (%). Normality was formally assessed using the Shapiro–Wilk’s test. Parametric continuous variables are presented as means and standard deviations (±SDs). Non-parametric variables are presented as medians and as first and third quartiles (Q1–Q3). Comparison between the groups for parametric continuous variables was performed using the independent samples Student’s *t* test. Kruskal–Wallis test was applied for non-parametric continuous variables. Differences between categorical variables was assessed using the chi-squared test. Type I error (alpha) was set at 0.05.

### 2.5. Data Analysis–Proteomics

Experimental data were processed using the Proteome Discoverer v2.4. software (Thermo Fisher Scientific, Waltham, MA, USA). MS spectra were searched against a human proteome database obtained from UniProt (Release 2024_01, Proteome ID: UP000005640) using SequestHT and Mascot search engines, with the following parameters: full trypsin protein digest and a maximum of 2 missed cleavages. Variable modifications were set for Met oxidation, Asn and Gln deamination and Met loss and/or acetylation for variable N-terminal modifications. Relative peptide/protein abundances were estimated using the label-free quantification (LFQ) algorithm, i.e., calculated from MS1 peptide intensities and normalized between samples. Individual patient samples were analyzed in technical duplicates. Obtained data were deposited at the ProteomeXchange Consortium via the Proteomics Identification Database (PRIDE) partner repository with the dataset identifier PXD054887.

Biological replicates (patient samples) were grouped by condition (fracture and non-union groups) and LFQ intensities were log-transformed. Protein group entries with less than 70% valid values of log-transformed LFQ intensity in each group were removed from further analysis [21]. Differentially expressed proteins (DEPs), i.e., proteins with a twofold ratio of overexpression or underexpression between the experimental groups, were determined using a both-sided, two-sample *t* test, with *p* < 0.05 results designated as statistically significant. Log_2_-transformed ratio of non-unions vs. fracture and log_10_-transformed *p*-values were displayed on a volcano plot [22]. Gene enrichment analysis of the identified statistically significant biological processes was performed using FunRich 3.1.4. software [23]. Statistical significance among the enriched biological processes was considered to be *p* < 0.05 by performing a hypergeometric test, corrected by Bonferroni for multiple comparisons. A comprehensive literature search was performed in order to verify the analysis and single out the processes considered most biologically relevant.

### 2.6. Data Analysis–Metabolomics

The filtering, peak detection, alignment, and integration were performed using Agilent MassHunter 10.1 software (Agilent Technologies, Waldbronn, Germany) [20]. For LC-MS data, the Molecular Feature Extraction (MFE) algorithm was used for deconvolution, followed by a second deconvolution step using the Recursive Feature Extraction (RFE) algorithm. Statistically significant accurate masses were analyzed with the CEU Mass Mediator [24] search tool in order to assign tentative metabolite candidates. For the deconvolution and identification of the GC-MS data, Fiehn library (version 2013, UC Davis, Davis, CA, USA) and NIST library (library 2.2 version 2014, Gaithersburg, MD, USA) were used. Multivariate statistical analyses were performed using Metaboanalyst version 6.0 (https://www.metaboanalyst.ca/ (accessed on 1 August 2024)) [25]. The variable importance was estimated using autoscaled data to a partial least squares-discriminate analysis (PLS-DA). Variable importance in projection (VIP) scores were obtained based on the PLS-DA models. The MetaboAnalyst 6.0 software was used for metabolomics data pathway analysis.

## 3. Results

Patients were categorized into two groups: those with normally healing fractures (N = 12) and those with pathological healing resulting in non-union formation (N = 11). Participants’ characteristics are presented in Table 1. The study outline and patient groups are depicted in Figure 1. The exclusion criteria were patients with severe systemic inflammation, malignant diseases, active infections, those who are immunocompromised, and those diagnosed with osteoporosis.

### 3.1. Proteomic Analysis and Gene Enrichment

A total of 301 and 289 proteins were isolated from the fracture and non-union groups, respectively. Proteins isolated from the two groups were mostly overlapping, as 288 proteins were identified in both groups, with 13 outlier proteins identified only in the fracture group, and only 1 in the non-union group, as is shown by a Venn diagram and table in Figure 2.

Gene enrichment analysis of biological processes showed three statically significant processes in the fracture group: cell growth and/or maintenance, protein metabolism, and immune response. The latter two are also shared by the non-union group; however, in this group, the “transport” biological process was also significant, while cell growth and maintenance was not. In both the fracture and non-union groups, the biological process of immune response was associated with the largest proportion of identified proteins, 21.4% and 22.7%, respectively (Figure 3).

An analysis of the DEPs led to the identification of six significantly upregulated proteins in non-unions vs. fractures: immunoglobulin heavy variable 3–74, immunoglobulin lambda variable 2–18, immunoglobulin lambda variable 2–11, low-density lipoprotein receptor-related protein 4, zinc-alpha-2-glycoprotein, and serum amyloid a-1 protein; and one downregulated protein was identified: cystatin-C, as shown by a volcano plot in Figure 4.

### 3.2. Metabolomic Analysis

PLS-DA analyses of metabolome data obtained by both the LC-ESI-QTOF-MS (in positive and negative analysis modes) and GC-EI-Q-MS analytical platforms revealed distinct metabolomics signatures between patients with bone fractures and patients with non-unions (Figure 5A–C, left panel). Top VIPs, i.e., metabolites that contribute the most to the model’s ability to discriminate between the two examined groups, are shown on the right panels in Figure 5A–C. The tentative identification of the top VIPs derived from PLS-DA analyses of LC-ESI-QTOF-MS metabolome data (Appendix A) suggests perturbations in the lipid metabolism, in particular, glycerophospholipids and sterol lipids. Analyses of data derived by GC-EI-Q-MS analyses identified nine VIPs above 1.7 (Appendix A). The metabolites that contributed the most to the difference between patients with bone fractures and non-unions were iminodiacetic acid, citric acid, 4-aminobutanoic acid, 3-hydroxyflavone, linoleic acid, N-methyl-DL-glutamic acid, 2-aminoadipic acid, aspartic acid, and 2-aminomalonic acid. In addition, N-methyl-DL-glutamic acid and 2-aminoadipic acid were identified in 10 (83%) of fracture patients while in only 4 (36%) non-union patients. To identify the pathways that could explain the differences between the two groups, we performed pathway analysis using MetaboAnalyst 6.0, which revealed the pathway of alanine, aspartate, and glutamate metabolism as significantly different (*p* = 0.000266, FDR = 0.0213).

## 4. Discussion

Although there is some debate on the diagnostic standards and timelines, non-unions are bone fractures that fail to heal over a prolonged period of time (usually defined at over 9 months), or show no visible healing progress (in 3 months). Pathophysiological mechanisms which lead towards “healing failure” are not completely understood; however, it is known that a plethora of both local (fracture type, blood supply, possible infection) and systemic factors (age, comorbidities, adequate nutrition) play a role in adequate bone healing [26]. Therefore, in order to elucidate potential systemic differences between the processes of active bone healing and the state of healing failure, we conducted a concomitant proteomic and metabolomic analysis of blood plasma in patients with normally healing fractures and patients with developed non-unions.

Our gene enrichment analysis identified that “protein metabolism” and “immune response” seem to be important biological processes in both patients with normally healing bone fractures and in non-union patients. The immune system plays a vital role in normal bone healing, especially in the inflammatory phase, by regulating the relationship between osteoblasts and osteoclasts [27,28]. Immune system dysregulation (most commonly clinically observed in acute and chronic inflammatory states such as polytrauma or diabetes mellitus, respectively) can significantly impair normal bone fracture healing [27]. Our proteomic analysis identified two variable (V) immunoglobulin protein domains (heavy variable 3–74 and lambda variable 2–18) to be significantly differentially expressed in the non-union group. Namely, variable (V) domains are important parts of immunoglobulin macromolecule, vital for the process of “V(D)J recombination”, which occurs in developing lymphocytes and leads to a vast antibody and T-cell receptor diversity; V domains participate in the antigen recognition process [29,30,31]. The role of immunoglobulins has been studied in the context of bone loss, mostly in patients with monoclonal immunoglobulin production (multiple myeloma), where it was found that these patients with elevated IgG levels have induced osteoclast function [28,32]. Activated IgGs enhance RANKL-mediated osteoclastogenesis. Additionally, increased trabecular bone mineral density has been observed in muMT mice which lack IgG, IgM, and the majority of IgA and B cells [33]. Additionally, it has been recognized that in the setting of rheumatoid arthritis, autoantibodies and the deposition of activated IgGs in the joint stimulate bone loss [28]. It is possible that specific immunoglobulins also play an important role in halting normal bone healing. Additionally, as expected, only patients with recent bone fractures (and presumed active healing) exhibited a significant expression of proteins related to “cell-growth and maintenance”, potentially reflecting the ongoing active tissue repair processes. Conversely, patients with non-unions exhibited a significant expression of proteins related to “transport”, indicating possible issues with ion transport, protein sorting and secretion, nutrient uptake, or waste disposal. The elevation of these mediators needed for proper mineralization and healing might point to issues in ensuring the proper ion balance and nutrient delivery to the fracture site in the non-union group. Furthermore, membrane protein transport dysregulation also impairs osteoclast function, resulting in an imbalance that can hinder proper healing [34].

Dysregulated transport processes can also affect lipid metabolism, leading to disruption in bone microenvironment homeostasis. Cholesterol abnormalities have been found to downregulate the Wnt pathway [35,36], one of the most studied signaling pathways involved in bone fracture healing. Activation of Wnt signaling is deemed helpful in the acceleration of bone repair [37,38]. We found low-density lipoprotein-related receptor 4 (LRP4) to be overabundant in the non-union group, when compared to the normally healing fracture group. Namely, LRP4 (along with other LDL-receptor-related proteins) has been recognized to regulate bone homeostasis through the Wnt pathway; for example, the binding of sclerostin to its canonical receptors (LRP5 or LRP6) leads towards the inhibition of Wnt-mediated bone formation induction. LRP4, recognized as a co-receptor of sclerostin, serves as an anchor that facilitates the binding of LRP6 and thus aids in Wnt pathway inhibition [39,40,41]. It is possible that LRP4 might be important in sustaining fracture healing failure in non-unions, especially when considering the fact that LRP5 mutations in mice have been shown to delay bone repair [38]. Another protein involved in lipid metabolism, zinc-alpha-2-gylcoprotein (ZAG), a known adipokine, was found to be overabundant in the non-union group. Namely, ZAG has been associated with lipolysis stimulation, the inhibition of lipid accumulation in adipose tissue, and the regulation of other adiopkines [42]. Due to the fact that its structure resembles the major histocompatibility complex (MHC)-I, it is likely that it also plays a role in immune system response [43]. Additionally, ZAG has been recognized as a possible biomarker of frailty and to be involved with tumor cachexia [42,44]. To the best of our knowledge, ZAG has not been directly implicated with bone fracture healing, but considering its still incompletely understood roles in lipid metabolism, it is plausible that it might indirectly impair bone fracture healing. This finding presents a new avenue to be explored.

Serum amyloid A1 is an apolipoprotein that we found to be overabundant in the non-union group, whereby in the acute phase setting, leads towards high-density lipoprotein (HDL) remodeling by displacing apoA-1. It has been postulated that SAA binding to HDL might lead towards the increased binding of HDL to macrophages, which aids in the removal of cholesterol from sites of inflammation [45,46,47]. Animal studies have shown that SAA1 likely blocks PTH-stimulated osteogenesis, thereby inhibiting the anabolic effect of PTH [48]. Finally, the lowered expression levels of a bone resorption inhibitor cystatin C in the non-union patients might additionally undermine the healing process [49].

We identified cholesteryl ester transfer protein (CETP) in the non-union group, but not in the fracture group. CETP plays a significant role in lipoprotein metabolism by facilitating the transfer of esterified cholesterol from HDL to lipoproteins and LDL in exchange for triglycerides [50]. This also points to possible perturbations of lipid metabolism in non-unions.

Our metabolomic analysis also identified differences between the fracture and non-union groups, which are primarily related to lipid metabolism, in particular, glycerophospholipids and sterol lipids. Namely, the top two VIPs identified by GC-MS analysis (linoleic acid and trans-13-octadecenoic acid) are both fatty acids, and the majority of top VIPs (identified tentatively by LC-MS) belong to categories of lipids (e.g., sterol lipids, glycerophospholipids, fatty acids). Further analysis has shown that the differences between the groups concerning their metabolomic profiles is due to significant differences in the pathway of alanine, aspartate, and glutamate metabolism. This is in line with several studies showing links between bone diseases and metabolic disorders affected by abnormal amino acid metabolism, as it is crucial for bone development and homeostasis [51]. Alanine, aspartate, and glutamate metabolism have all been linked to lipid metabolism, most specifically in the setting of metabolic syndrome [52,53]. Other researchers, like Liu et al., also showed that patients with femoral necrosis and fractures also present with major metabolites involved in glycerophospholipid metabolism and alanine, aspartate, and glutamate metabolism [54]. Whether these metabolomic changes are the cause of dysfunctional bone metabolism, its consequence, or reflections of patient’s risk factors for non-union formation related to their comorbidities, remains to be further studied.

Our cross-sectional concomitant proteomic and metabolomic study has several limitations. Firstly, plasma samples were pooled for metabolomic analysis. Another limitation is the small sample size. Additionally, this study had fewer female than male participants. A potential source of proteomic and metabolomic bias might be the fact that some normally healing fracture patients underwent surgery prior to a sample being obtained. In addition, we do not know if some of our patients from a recent fracture group might develop non-unions later in the future. Another important fact is to consider the potential confounders which were not controlled, such as a larger age range in the non-union group, but also factors which were not assessed, including physical activity, nutritional status, etc.

In spite of the above-mentioned limitations, our pilot study is, to the best of our knowledge, the first to comprise a concomitant metabolomic and proteomic analysis in the field of non-union research. By conducting these two analyses, the identified DEPs related to lipid metabolism may have opened a new horizon in non-union research. Additionally, the likely role of immune system regulation has been strengthened. It is important to consider the limitations of our study when drawing conclusions and take into account that further focused research is required to validate or dismiss our findings.

## 5. Conclusions

Non-unions are chronic bone fractures that fail to heal, with unclear underlying causes. Therefore, we conducted a cross-sectional proteomics and metabolomics pilot study that investigated the mechanisms driving this condition. Our findings suggest that inflammation and disrupted protein transport lead to imbalances in lipid metabolism, ultimately compromising the bone healing process.

## Figures and Tables

**Figure 1 bioengineering-11-01135-f001:**
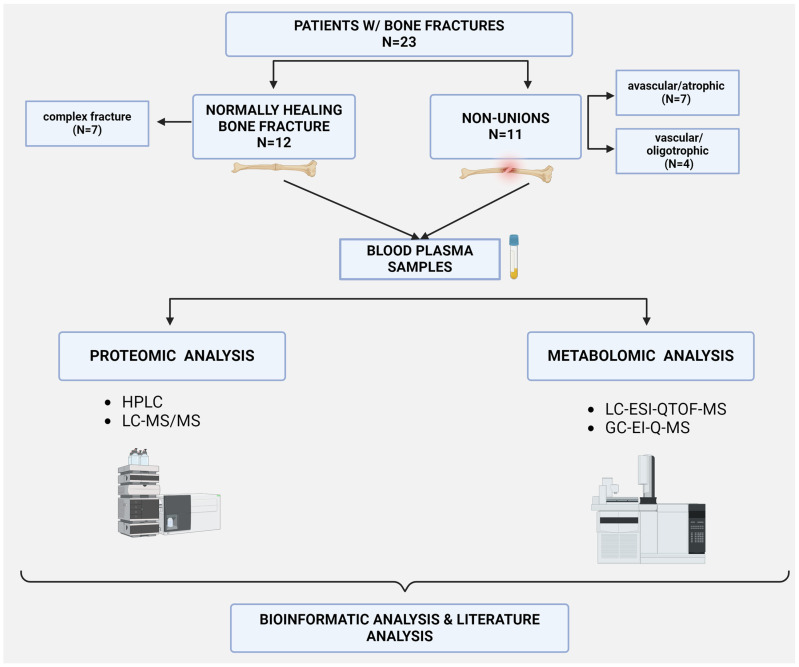
Study outline depicting patient groups and methods of concomitant proteomic and metabolomic sample as well as data analysis. Image created with Biorender.com.

**Figure 2 bioengineering-11-01135-f002:**
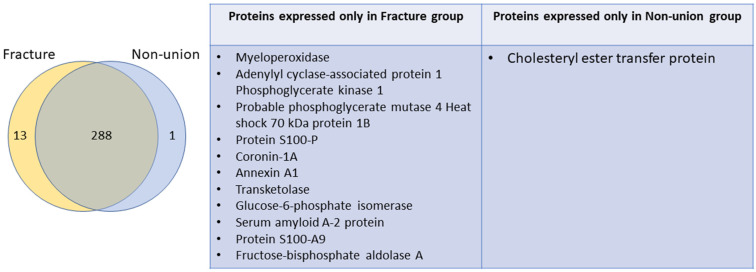
Venn diagram (**left**) showing identified proteins in both groups: fracture and non-union. Overlapping proteins (288) were identified in both groups, while 13 and 1 outlier proteins were identified in the fracture and non-union group, respectively. The proteins identified only in one group are listed in the table (**right**).

**Figure 3 bioengineering-11-01135-f003:**
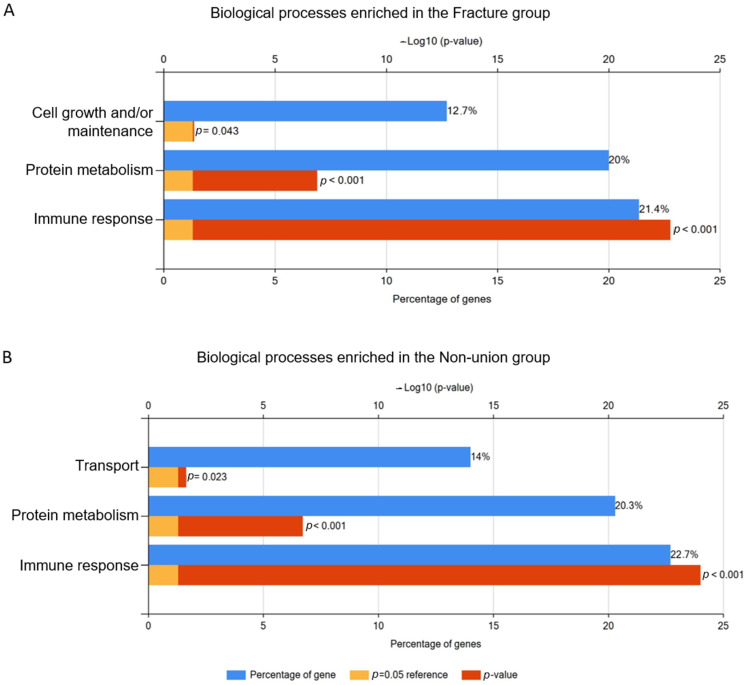
Bar chart showing biological processes which were significantly associated with the protein set identified in the bone fracture (**A**) and non-union (**B**) group. Blue columns show percentage of identified proteins associated with each respective process, red and orange bars show *p* value. Created with FunRich 3.1.4. software.

**Figure 4 bioengineering-11-01135-f004:**
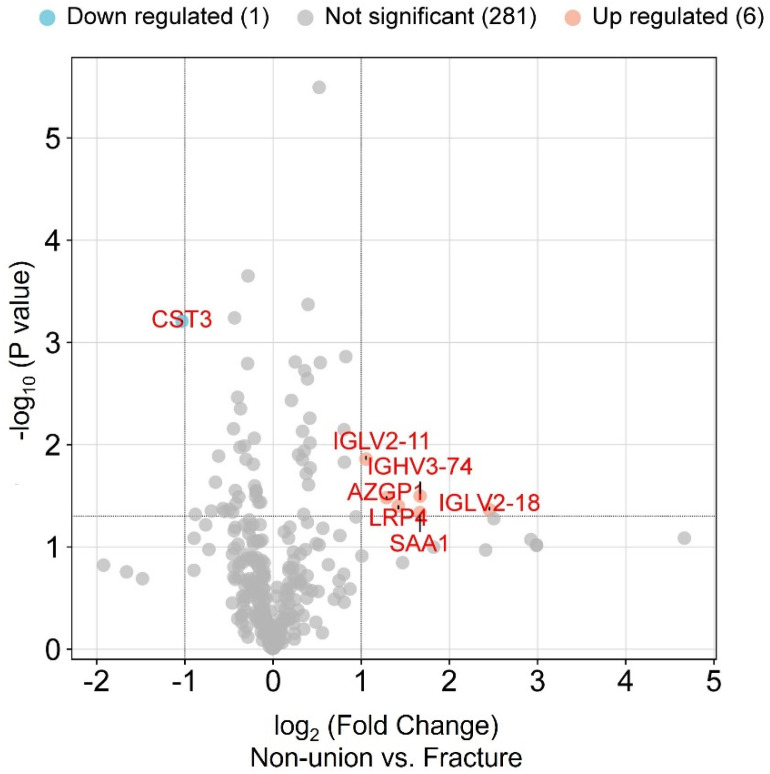
Volcano plot comparing protein expression levels in the two groups: bone fracture and non-union. Up- and downregulated proteins are shown in orange and blue circles, respectively. Proteins with (i) *p*-value ≤ 0.05 determined by a both-sided, two-sample t-test and (ii) mean log-transformed ratio non-union vs. fracture higher than 1 or lower than −1 were termed as significantly differentially expressed proteins. The mentioned thresholds are indicated by bolded lines on the graph. Proteins found to be significantly overexpressed in the non-union group are immunoglobulin heavy variable 3–74 (IGHV3-74), immunoglobulin lambda variable 2–18 (IGLV2-18), low-density lipoprotein receptor-related protein 4 (lrp4), zinc-alpha-2-glycoprotein (azgp1), and serum amyloid a-1 protein (saa1), while significantly lower expression was shown by cystatin-C (CST3).

**Figure 5 bioengineering-11-01135-f005:**
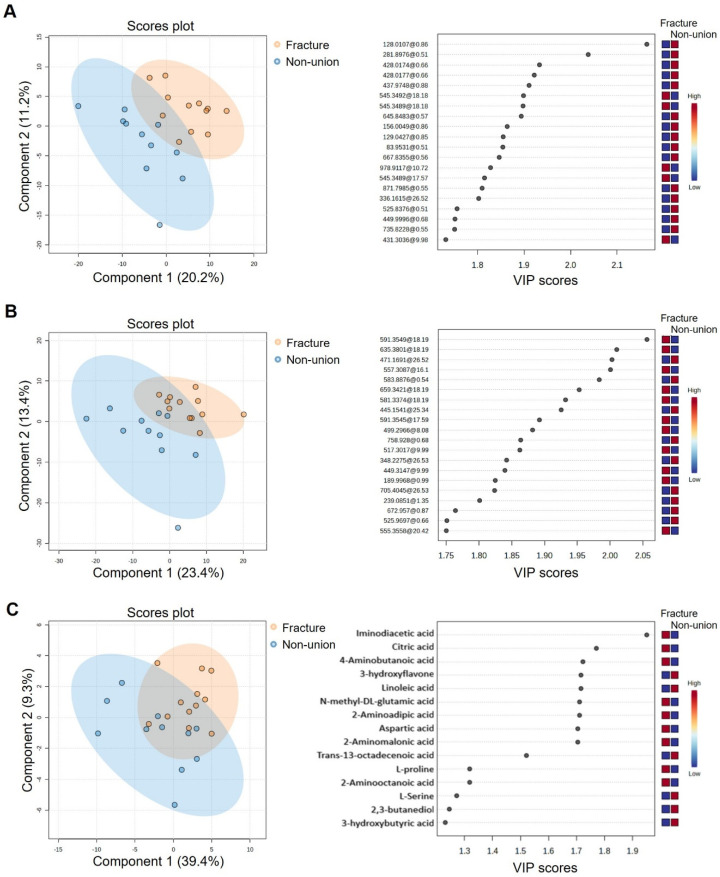
Differences in metabolite expression in the fracture vs. non-union group detected by three Partial least squares-discriminate analysis (PLS-DA) score plots and variable importance in projection (VIP) scores of metabolomics data obtained by liquid chromatography–electrospray ionization–quadrupole time-of-flight mass spectrometry (LC-ESI-QTOF-MS) and gas chromatography–electron ionization–quadrupole mass spectrometry (GC-EI-Q-MS) analytical platforms. Expression levels of analytes are shown on heatmaps (**A**) LC-ESI (+) QTOF data, (**B**) LC-ESI (−) QTOF data, and (**C**) GC-MSD data.

**Table 1 bioengineering-11-01135-t001:** Participants’ general characteristics.

	Physiological Fracture Healing	Non-Union	Statistical Difference
Number of participants N (%)	12	11	
Age (years): Mean (±SD) Range	40.7 (±7.2) 27–50	40.5 (±15.8) 22–69	*p* = 0.967 (t = 0.042)
Female sex: N (%)	1 (8.3%) (47 years old)	1 (9.1%) (69 years old)	*p* = 0.949 (χ^2^ = 0.004)
Fracture/non-union site * (N)	ulna (5), radius (5), tibia (1), clavicle (1), scapula (1), humerus (1), fibula (2), tibia (1)	scaphoid (5), femur (5), tibia (1), fibula (1), clavicle (1)	
Non-union type: N (%)	N/A	avascular/atrophic: 7 (63.6%) vascular/oligotrophic 4 (36.4%)	
Treatment type N (%)	conservative: 7 (58%) surgical: 5 (42%)	conservative: 0 surgical: 11 (100%)	*p* = 0.002 (χ^2^ = 9.22)
Days from fracture onset to blood draw: median (Q1–Q3)	13.5 (8–16)	360 (195–825)	*p* < 0.001 (χ^2^ = 16.5)
Smoking: N (%)	4 (3.3%)	4 (36.4%)	*p* = 0.879 (χ^2^ = 0.023)
NSAID use	3 (25%)	1 (9%)	*p* = 0.315 (χ^2^ = 1.01)
PPI use	1 (8.3%)	2 (18.2%)	*p* = 0.484 (χ^2^ = 0.491)
CCI index median (Q1–Q3)	0 (0–0)	0 (0–0.75)	*p* = 0.151 (χ^2^ = 2.06)

CCI—Charlson comorbidity index; PPI—proton pump inhibitors. * Several participants had multiple fracture/non-union sites.

## Data Availability

Obtained proteomics-based data were deposited at the ProteomeXchange Consortium via the Proteomics Identification Database (PRIDE) partner repository with the dataset identifier PXD054887.

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
