# Peer review of "Systemic Lipid Metabolism Dysregulation as a Possible Driving Force of Fracture Non-Unions?"

_bioengineering, 2024, doi:10.3390/bioengineering11111135_

Round 1

Reviewer 1 Report

Comments and Suggestions for Authors

In this study, the author investigated “Systemic Lipid Metabolism Dysregulation as a Possible Driving Force of Fracture Non-unions?”. After reading this content, the findings are interesting and presented well. The manuscript can be accepted for publication after minor corrections.

Some critical suggestions

1.      Keywords are redundant. Please correct it.

2.      In this study, proteomic data analysis related to blood plasma depends on spectrometry instead of using molecular techniques. Why not perform?

3.      Why not consider bone osteogenic proteins?

4.      Although, the authors state the limitations of the study. What about the status of immunohistochemistry of skeletal tissues?

5.      Please provide the conclusion separately.

Author Response

Reviewer 1:

In this study, the author investigated “Systemic Lipid Metabolism Dysregulation as a Possible Driving Force of Fracture Non-unions?”. After reading this content, the findings are interesting and presented well. The manuscript can be accepted for publication after minor corrections.

Some critical suggestions

  1. Keywords are redundant. Please correct it.

We agree, the „omics“ and „multiomics“ are indeed redundant and were removed as such. The keywords now list: bone healing, bone fracture, non-union, proteomics and metabolomics.

  1. In this study, proteomic data analysis related to blood plasma depends on spectrometry instead of using molecular techniques. Why not perform?

We thank the Reviewer for their comment. Indeed, it is common to use LC-MS in pilot investigations such as ours, as MS provides an untargeted and highly specific identification and relative quantification of a large number of target proteins without the need for antibodies.

This offers a broad and un-biased initial screening for tentative biomarker candidates. Although the scope of this research was rather narrow, a larger patient cohort using quantitative methods is planned: As the key molecules were identified, immunological methods will be used in future research for a more targeted and cost-effective analysis in larger sample sets.

  1. Why not consider bone osteogenic proteins?

Thank you for the insightful suggestion. We have considered the study of osteogenic proteins, however our results in this study have not pointed towards significant or reportable expression of these proteins. Our presumption is that this is due to the fact that bone morphogenetic proteins are mostly active during the early stages of bone healing. Their roles in bone repair are therefore crucial for initiating and further regulating the complex early cascades of molecular events that lead to bone regeneration. As such, they are less likely to be identified in our samples from patients with bone non-unions, as normal bone healing, i.e. osteogenesis has failed to occur, and 3 to 6 months has passed since the original fracture. Likely, the osteogenic proteins do not play a key role in the mechanism of this condition. Instead, our results point to the underlying inflammatory processes leading to lipid dysregulation as an important mechanism of bone non-unions. This is in line with other published reports on bone pathophysilology (i.e. pathological bone formation) where ephrin B mediated inflammation was center stage in fibrodiysplasia ossificans progressiva patients (Hrkac S et al. Bone Rep. 2022 Feb 28;16:101177. doi: 10.1016/j.bonr.2022.101177).

It must also be considered that osteogenic proteins only occur scarcely in peripheral circulation, which is likely an additional reason why they were absent in our analysis. However, the question points to a new avenue of proteomic studies of bone metabolism – which is analysis of intraoperatively taken tissue samples of pathological (non-union) fractures. As these proteins are likely to be more abundant locally, this should provide a better insight into the osteogenic mechanisms and the profiles of the involved proteins that are “diluted” and thus challenging to obtain from peripheral blood samples.

  1. Although, the authors state the limitations of the study. What about the status of immunohistochemistry of skeletal tissues?

We thank the Reviewer for their comment. The aim of our study was to assess proteomic and metabolomics systemic factors, which might influence/reflect abnormal bone healing. Therefore, we chose blood plasma.

Additionally, as explained in question #3, we agree that investigating the bone tissues would in theory provide a more precise insight into the molecular mechanisms comparing normal bone healing and non-unions. However, this sample was not available to us.

  1. Please provide the conclusion separately.

We have added the Conclusion section to the bottom of the Discussion part of the manuscript that reads: „Non-unions are chronic bone fractures that fail to heal, with unclear underlying causes. Therefore, we conducted a cross-sectional proteomics and metabolomics pilot study that investigated the mechanisms driving this condition. Our findings suggest that inflammation and disrupted protein transport lead to imbalances in lipid metabolism, ultimately compromising the bone healing process.“

Reviewer 2 Report

Comments and Suggestions for Authors

In the manuscript “Systemic Lipid Metabolism Dysregulation as a Possible Driving Force of Fracture Non-unions? Grgurević  et al. presented a very interesting proteomic and metabolomic analysis of blood plasma in patients with normally healing fractures and patients with developed non-unions.

The objectives and results are thoroughly presented. However, even though the results are interesting they remain indicative without the use of quantitative analyses (e.g. qPCR, RNAseq) in a larger pool of patients. Besides the small cohort size, results could be also largely affected by factors, like age, nutrition and physical activity, known to influence the process of bone healing. Specifically, the 19 years difference in the age range between the two study groups could be a significant modifier, especially if the non-union group is composed of outliers. As a matter of fact, the double SD for the, otherwise, similar mean age, between the two groups could be explained by either the presence of age outliers or a widespread age distribution. Moreover, the metabolic turnover rate is influenced by aging, exercising and nutritional habits. So, besides the age range, it is important to verify that the study groups have similar characteristics regarding these attributes.  

So, is it possible to a) verify the presence of age outliers (e.g. above 60), and/or b) test whether the age range is statistically significant between the two groups? Also determine the levels of different proteins and genes using perhaps multivariable logistic regression analysis after adjustment for co-founders, including age, gender, types of NSAIDs used, NSAIDs taking duration time, type of conservative treatment used, etc.  Finally, is the only female for each group menopausal or not?

However, the results are interesting and should be followed using a larger cohort.

Author Response

Reviewer 2

In the manuscript “Systemic Lipid Metabolism Dysregulation as a Possible Driving Force of Fracture Non-unions?” Grgurević  et al. presented a very interesting proteomic and metabolomic analysis of blood plasma in patients with normally healing fractures and patients with developed non-unions. The objectives and results are thoroughly presented. However, even though the results are interesting they remain indicative without the use of quantitative analyses (e.g. qPCR, RNAseq) in a larger pool of patients. Besides the small cohort size, results could be also largely affected by factors, like age, nutrition and physical activity, known to influence the process of bone healing.

We thank and agree with the Reviewer on their important remarks. We fully acknowledge the limitations of our study, which we now further highlighted by expanding the limitations section of our discussion (cited below).

Specifically, the 19 years difference in the age range between the two study groups could be a significant modifier, especially if the non-union group is composed of outliers. As a matter of fact, the double SD for the, otherwise, similar mean age, between the two groups could be explained by either the presence of age outliers or a widespread age distribution. Moreover, the metabolic turnover rate is influenced by aging, exercising and nutritional habits. So, besides the age range, it is important to verify that the study groups have similar characteristics regarding these attributes.  

We thank the Reviewer for pointing this out, prompted by the Reviewer's remark; we have completely revised Table 1, in order to include a formal statistical assessment of the participants' characteristics. Details regarding statistical analysis of these parameters are now included in the new materials and methods subsection. Namely, in spite of an increased range, we did not find a statistically significant difference in the age group. However, we further discussed this in the limitations section, which now states:

„Another important fact is to consider the potential confounders which were not controlled, such as a larger age range in the non-union group, but also factors which were not assessed, including physical activity, nutritional status, etc. “

So, is it possible to a) verify the presence of age outliers (e.g. above 60), and/or b) test whether the age range is statistically significant between the two groups?

We thank the Reviewer; we have performed a formal statistical assessment and presented the results in the (now revised) Table 1. We further addressed this in the Discussion section (Cited above)

Also, determine the levels of different proteins and genes using perhaps multivariable logistic regression analysis after adjustment for co-founders, including age, gender, types of NSAIDs used, NSAIDs taking duration time, type of conservative treatment used, etc.  

We thank the Reviewer for their suggestion. Namely, as this work included a concomitant proteomic and metabolomic shotgun analysis in a rather small cohort of patients, we do not find it feasible to perform such an elaborate statistical analysis due to the following reasons: As the sample size is quite small, a multivariable logistic regression analysis for each protein/metabolite consisting of so many covariates would likely lead to model overfitting and possibly to erroneous assumptions due to multiple testing. Furthermore, as the Reviewer stated, the fact that the LC-MS is semi-quantitative in nature, we believe that such analyses would not yield applicable conclusions. 

However, this suggestion would be feasible and very useful in selected proteins determined using a quantitative method (e.g. ELISA) on a larger cohort, which we plan on performing in future research projects based on this pilot study.

Finally, is the only female for each group menopausal or not?

We thank the Reviewer for their question: The female participant from the fracture and the participant from the non-union group were 47 (not menopausal) and 69 years old (underwent menopause), respectively. This has now been included in the Table 1.

However, the results are interesting and should be followed using a larger cohort.

We thank the Reviewer. We plan to perform a more focused research using quantitative methods of selected proteins and metabolites on a larger cohort.